# Survey on Psychosocial Conditions of Official Veterinarians in Germany: Comparison with Other Professions and Differences between Age Groups, Gender, and Workplace Characteristics

**DOI:** 10.3390/ani14131975

**Published:** 2024-07-03

**Authors:** Katharina Charlotte Jensen, Roswitha Merle

**Affiliations:** Institute of Veterinary Epidemiology and Biostatistics, Center for Veterinary Public Health, School of Veterinary Medicine, Freie Universität Berlin, 14163 Berlin, Germany; roswitha.merle@fu-berlin.de

**Keywords:** mental health, COPSOQ III, veterinary authorities, survey, welfare controls, job satisfaction, psychosocial conditions, One Health, wellbeing

## Abstract

**Simple Summary:**

Veterinarians often suffer from mental diseases. Related studies have mostly regarded veterinarians working in practice, so little is known about veterinarians working for authorities (official veterinarians). Official veterinarians control animal husbandries or food producers to ensure animal welfare, freedom from infectious diseases, and food safety. We assessed the psychosocial wellbeing of official German veterinarians and compared it to that of employees in other professions using an online questionnaire. The results indicated that official veterinarians experience high demands during their work and are regularly exposed to offensive behavior during their activities. They also felt exhausted more often than employees in other professions. They also considered their work as meaningful and had more freedom than other employees concerning working time and how they performed their jobs. The results indicate that official veterinarians may require protection and support.

**Abstract:**

Background: Even though the mental health of veterinarians has raised concerns, little is known about the wellbeing of official veterinarians ensuring animal welfare, food safety, and control of infectious diseases and performing other tasks for veterinary public health. Therefore, the aim of this study was to assess the psychosocial conditions of official German veterinarians and to compare them with those of other employees. Methods: An online survey was conducted including 82 items of the Copenhagen Psychosocial Questionnaire (COPSOQ III) and additional questions on workplace and demographics. Results: The answers of 838 respondents were analyzed, representing 26% of the target population. The average values for the dimensions *Quantitative Demands*, *Emotional Demands*, and *Burnout Symptoms* were substantially higher than those for German employees in other professions. Official veterinarians mainly working in animal welfare frequently experienced physical violence (6.7%) or threats of violence (53%). However, the profession also provides advantages: official veterinarians reported higher average values concerning the dimensions *Control over Working Time* and *Influence at Work* than other employees. Most participants stated that they experienced their work as meaningful, and the *Job Insecurity* dimension was low. Conclusions: Official veterinarians frequently experience offensive behavior and high-level demands. Therefore, measures to support and protect official veterinarians are needed. The positive aspects of this job should be emphasized to recruit and retain veterinarians in this field of the profession.

## 1. Introduction

The mental wellbeing of veterinarians is a cause for concern. Veterinarians more often suffer from stress and depression and are more likely to commit suicide than the general population [1,2]. This phenomenon has been observed for several decades worldwide [3,4,5,6,7,8,9]. However, the awareness, discussion, and research of this topic have focused on veterinarians working as practitioners. Concerning veterinarians working for the authorities in cities, districts, federal states, or federal ministries (hereafter referred to as official veterinarians), little is known. Additionally, official veterinarians might suffer from work-related mental health problems; for instance, two official veterinarians committed suicide in Germany in 2015, one after being threatened by farmers [10].

Official veterinarians provide several different functions in the context of One Health such as controlling compliance with laws and other regulations. In Germany, they ensure the health of humans by inspecting food of animal origin in slaughterhouses as well as in shops or restaurants, monitor infectious diseases in livestock and pets, and inspect households and farms to ensure animal welfare. Moreover, official veterinarians inspect or control animal by-products, animal experiments, veterinary pharmacies, and the trade of animals and food of animal origin at the border. In 2021, 10% of all veterinarians (3224 of 32,930) were employed as official veterinarians in Germany [11]. Worldwide, no current information is available on the field of work of 10 million veterinarians [12]. In 2003, approximately 25% of veterinarians worked for the government (also including veterinarians performing animal health services funded by the government [13]).

Several risk factors have been identified regarding the mental health of veterinarians working in practice. For instance, dealing with bereaved clients [14], long working hours [15], and performing euthanasia [7] are associated with distress, anxiety, and/or sadness. However, the working life of official veterinarians is different from that of practicing veterinarians; therefore, these factors may not fully apply to official veterinarians. For instance, official veterinarians do normally not perform euthanasia but may have to order and/or perform the culling of thousands of animals to control infectious diseases. In these as well as in other situations, official veterinarians are highly accountable and must weigh different interests and aspects. Unannounced controls concerning animal welfare or food safety are part of the everyday work for most official veterinarians, which can be stressful [16] as the situations are unforeseen. The persons facing these controls are often stressed and sometimes ready to use violence, such as when pet owners fear that their pet will be taken away or when farmers fear losing their livestock, which are their source of income and a part of their identity. In Germany, several attacks on official veterinarians have recently been reported. The most severe cases were caused by farmers: an official veterinarian was killed in 2015, two official veterinarians were injured with an iron rod in 2016, and an official veterinarian was shot in 2017 [17]. Official veterinarians often have to balance the interests of animals and humans, which can lead to moral distress [18]. Animal welfare incidents are often associated with the social, health, and/or psychological difficulties of the owners. Determining an adequate response is challenging for official veterinarians [19]. However, how strongly these factors influence the wellbeing of official veterinarians is unknown.

A second factor that might be demanding for official veterinarians is that all controls and conversations have to be documented to enable their use as proof in court proceedings. In addition, workload impacts the wellbeing of veterinarians as well as other types of employees [20,21]. The work tasks of official veterinarians have probably increased in the recent years due to different factors: new regulations that have been enacted, such as the Animal Health Law in 2021 [22]; more emphasis being placed on animal welfare; and, finally, the prevalence of new or reoccurring infectious diseases, e.g., the highly pathogenic avian influenza and African swine fever. In Germany, the veterinarian workload also increased in official veterinary service due to skills shortages and job vacancies [23,24]. All these factors, including high levels of responsibility, conflicts, requirements for court-proofed documentation, moral distress, and high workload, have presumably resulted in high job demands. However, scientific research on the demands faced by official veterinarians in their jobs is limited. Worldwide, to the best of our knowledge, only one scientific study has been conducted on the wellbeing of official veterinarians [16]. Ultimately, knowledge must be obtained to keep as many official veterinarians in their jobs as possible and to attract more veterinarians to these positions to ensure food safety, animal health, and animal welfare [24]. Therefore, the structural benefits and demands the professionals experience must be understood. As such, the aim of this study was to assess the status of the demands, social relationships, leadership, influences, and potential at work; conflicts and violence; as well as the job satisfaction and (mental) health of official veterinarians using the Copenhagen Psychosocial Questionnaire (COPSOQ III). Where possible, the results were compared with those of employees in other professions to identify differences. In particular, we hypothesized that official veterinarians experience high levels of cognitive and emotional demands and often experience conflicts and offensive behavior during their work. Finally, we analyzed whether the emotional and cognitive demands as well as the burnout symptoms differ between age groups, genders, persons in leadership positions, and different fields of work (animal welfare or others).

## 2. Materials and Methods

The questionnaire used in this study was developed from the third version of the Copenhagen Psychosocial Questionnaire (COPSOQ III) described by Burr et al. [25]. This questionnaire is a generic instrument used to assess the different dimensions of psychosocial conditions and is applicable for all kinds of jobs [25]. The COPSOQ III is not a ready-to-use questionnaire but a catalog of questions that are grouped into three different levels: core, middle, and long. For this study, all 31 core, 13 middle-level, and 38 long-level questions were used (Appendix A). The questions belonged to the different dimensions of psychosocial conditions, e.g., *Job Satisfaction* and *Emotional Demands* [25]. The 82 questions used in this version were from 40 different dimensions. We used most of the items and the phrasing of the validated German Standard Version (GSV [26]). The GSV questions were supplemented using questions from the COPSOQ III that were considered relevant for the purposes of our study. In particular, questions relating to conflicts and offensive behavior were added. Participants were asked about their experiences in the last 12 months concerning gossip and slander, conflicts and quarrels, cyber bullying, sexual harassment, threats of violence, and physical violence. If a person reported that they experienced such behavior, they were asked by whom (owner of animals, food producer, coworker, supervisor, or someone else). Moreover, demographic information (gender, age, and federal state), information concerning professional experience and current workplace, and the availability and the attitude toward certain measures that might reduce work-related demands were requested. All questions except the open questions were mandatory. In total, the questionnaire comprised 139 questions and is available in Appendix A in the original language (German). The questionnaire was pretested by a colleague with expertise in psychology as well as by three official vets. Feedback was received either by email or voicemail. After the pretest, only minor changes were made.

This study’s design and questionnaire were approved by the Ethics Committee of the Freie Universität Berlin (ZEA-Nr.: 2022-024). Each participant was informed of the aims of the study and that some questions might evoke memories of demanding situations. They were further informed of the personal-related data such as gender that would be collected. The possibility of quitting the survey without further consequences was mentioned. IP addresses or other individual-related data were not collected. All participants actively agreed to participate. The data of the participants were treated confidentially.

The questionnaire was implemented in LimeSurvey Cloud and was accessible via an internet link (https://vetepi.limesurvey.net/935244?lang=de accessed on 11 June 2024). A short introduction informed the participants concerning the aim of the study and data protection measures. Questions were displayed on 12 pages. At the end of the survey, the participants had the opportunity to provide comments. The survey was available from 1 November 2022 to 23 of January 2023. The invitation to participate was shared on a Facebook group of official veterinarians and was published in the Journal of the Veterinary Chamber in Germany. Moreover, the association of official veterinarians of Germany (Bundesverband der beamteten Tierärzte e.V.) distributed the invitation via e-mail to their members.

The answers were exported to Microsoft Excel© (16.0) and imported to Statistical Analyses Software (SAS© Version 9.4), which was used for data handling as well as statistical analyses. Figures were created using Microsoft Excel© (16.0). The answers were checked for plausibility, where possible. The answers from persons who stated they did not work as an official veterinarian and those who did not complete at least half of the questionnaire were excluded.

All questions from the COPSOQ III were answered with a 5-point Likert scale, except the General Health item (1–10) and items of the *Self-Efficacy* dimension (4-point Likert scale). The categories were replaced with the equivalent numbers provided in the guideline (0-25-50-75-100 or 0-33-67-100) [24]. Then, the values of the dimensions were determined by calculating the mean over the corresponding items. For instance, for the *Role Conflicts* dimension, the mean of the answers to the questions “Are contradictory demands placed on you at work?” (item CO2; Appendix A) and “Do you sometimes have to do things which seem to be unnecessary?” (item CO3) was calculated. Corresponding to the items, all dimensions ranged between 0 and 100. If an item was missing, the dimension was set to missing. As the questionnaire used in this study also included questions other than the those in the GSV, six dimensions were calculated in two different ways: first, including all items that were used in our study; second, including only those items that were used in the GSV [26] to ensure comparability to the reference data. For example, in our study, the *Emotional Demands* dimension included items ED1, EDX2, and ED3. ED1 was not part of the GSV. Therefore, we first calculated the dimension with all three items and second with only EDX2 and ED3 as in the GSV. The descriptive analyses included the mean and its 95% confidence interval. The means and confidence intervals were compared with those of a reference population [26], including at least 148,435 German employees. Statistical tests could not be conducted as the data were not freely available. For some data, the median as well as 25% (Q1) and 75% (Q3) percentiles were also calculated. The dimensions had one to four corresponding items (Appendix A). Those dimensions consisting of only one item were not continuous as they could only contain the five different values. However, to ensure comparability, we followed the guideline but also reported the percentage of participants within each category. The Mann–Whitney U-test and Kruskal–Wallis test were used to identify differences between subgroups concerning different dimensions (e.g., gender and *Emotional Demands*). The associations between subgroups and categorical data (e.g., main field of work and experience with violence) were tested with chi square tests. *p*-values < 0.05 were regarded as statistically significant.

## 3. Results

### 3.1. Participants

Overall, 886 persons opened the survey and answered at least the first question. A total of 19 persons reported not working as an official veterinarian (defined as working for the government in the fields of animal welfare, food safety, infectious diseases, in the slaughterhouses, or having other responsibilities of authorities), and 28 persons did not finish at least half of the questionnaire. These persons as well as one other person who provided clearly wrong answers were excluded from further analyses. Therefore, the answers of 838 participants were analyzed. This represented 26% of the target population [11].

An average of 21 min was required to complete the questionnaire (median; Q1: 16 min; Q3: 30 min). The information on demographics and job characteristics is displayed in Table 1. Where possible, the percentage in the study was compared with that of the target population [11] or reference population [26]. Nearly 80% of participants were women, which is a higher proportion than in the target population. Two-thirds of the participants worked full-time. The percentage of civil servants was higher in the study population than in the target population, and 22% of the study participants were 55 years or older. The median work experience was 10 years (Q1: 4 years, Q3: 19 years). Most of the participants worked mainly in the field of animal welfare (Table 1). Approximately 41% (*n* = 348) of the participants reported performing a supervisory function for other official veterinarians.

### 3.2. Demands

The descriptive results of the dimensions are shown in Table 2, and the comparison with the reference population is displayed in Figure 1. Most participants stated that their job was highly demanding: concerning the *Cognitive Demands* dimension, most participants reported to often or always have to track and remember many things and that they often or always have to make difficult decisions. This resulted in a mean value of 82 for *Cognitive Demands* (Table 2). The mean score of 71 for *Emotional Demands* was substantially higher than that the reference population (48; Figure 1). Further analyses were conducted for these two dimensions to gain deeper insights. The score for *Emotional Demands* did not vary among genders or age groups, but veterinarians working in animal welfare reported experiencing significantly higher emotional demands (Table 3). Veterinarians in a leadership position for other veterinarians reported higher levels in the *Cognitive Demands* dimension (Table 3). The main field of work did not influence the degree of strain. 

### 3.3. Influences and Possibilities for Development

The assessments concerning the *Influence at Work* and *Possibilities for Development* dimensions were relatively positive, with mean scores of 56 and 74, respectively (Table 2). This result means that, on average, the veterinarians often had a strong influence on decisions, and they often could influence what they performed and how they did it. However, they had less of an influence on the amount of work assigned to them (item IN3): here, 23%, 44%, 26%, and 8% of the participants answered “never/hardly ever”, “seldom”, “sometimes”, and “often” or “always”, respectively. The *Control over Working Time* dimension was assessed higher by the study population than by the reference population (Figure 1). Moreover, the participants stated they worked less often at night and on the weekend than the reference population (Table 1). Additionally, the *Meaning of Work* dimension was positively assessed, as 41% and 28% of participants stated that they felt their work was meaningful “often” and “always”, respectively.

### 3.4. Social Relationships and Leadership

Most participants reported always receiving social support from their colleagues if needed and described the working atmosphere with their colleagues (*Sense of Community at Work* dimension) as good (Table 2). The responses for the *Quality of Leadership*, *Social Support by the Supervisor*, and *Recognition by the Management* dimensions varied more than those for the other scales (Table 2). This finding indicates that some participants were satisfied with their supervisor, while others were not. The scores concerning the *Role Clarity*, *Role Conflicts*, *Illegitimate Tasks*, and *Organizational Justice* dimensions varied less, around 50 points. This means that most participants agreed only partly to terms such as “Do you receive all the information you need in order to do your work well?” or “Does your work have clear objectives?” (*Role Clarity* dimension). However, the assessments of the study population concerning the *Recognition*, *Role Conflicts*, *Predictability*, and *Trust and Justice* dimensions resembled those of the reference population (Figure 1).

### 3.5. Job Security and Job Satisfaction

The participants reported feeling rather secure concerning their employment (Table 2). This finding can be explained by the high percentage of civil servants participating in the study (Table 1).

Most participants stated that they were “satisfied” or “neither satisfied nor unsatisfied” with the different aspects of their work, resulting in a mean score of 61 on the *Job Satisfaction* dimension (Table 2). The answers varied concerning the different job aspects: 70% chose the answers “satisfied” or “very satisfied” concerning their salary, but only 42% were “satisfied” or “very satisfied” with their management.

Regarding workplace commitment, the agreement with the item “Do you enjoy telling others about your place of work?” was lower than that with the other two items. The reason for this finding might be that, in some cases, the information is confidential or other persons do not like to hear stories about animal abuse, slaughter, or hygiene deficiencies.

### 3.6. (Mental) Health

All questions concerning the health of the participants related to their general wellbeing during the last four weeks, as health status could be assigned to only personal or only work-related reasons, so this must be kept in mind when interpreting our results.

The assessment of the general health of the study population was slightly lower than that of the reference population (Figure 1), which might have been related to the study population being older (Table 1). However, the study population also scored higher in the *Burnout* dimension (Figure 1). The average *Burnout* score of 58 indicated that most of the participants reported feeling worn out and physically and emotionally exhausted (most) part of the time. Between 7% and 10% of the participants answered these questions with “all the time”. The results of further analyses revealed that in contrast with cognitive and emotional demands, the *Burnout* assessment differed among genders and age groups but not concerning the job characteristics (Table 3). Women reported a higher level of *Burnout* symptoms than men, and persons in the age group 35–44 years reported a higher level of *Burnout* than those older than 54 years.

### 3.7. Conflicts and Offensive Behavior

More than half of the participants reported being involved in gossip and slander and more than 80% in conflicts and quarrels in the last 12 months (Figure 2). This was mostly the case with animal owners (61% and 68%) but also with colleagues (43% and 41%) or supervisors (20% and 24%; multiple choices possible).

A total of 19 persons reported that they had been sexually harassed by animal owners (*n* = 9), colleagues (n = 3), food producers (n = 2), supervisors (n = 2), and others (*n* = 3). Approximately one-third of participants (35%) reported experiencing threats of violence, 93% (*n* = 264 of 285) of them by animal owners and only rarely by food producers (*n* = 28) or others (*n* = 10). Veterinarians working mainly in the field of animal welfare were more likely to be threatened (53% vs. 21%, *p* < 0.001) than those working mainly in other fields. A total of 30 persons reported experiences with physical violence, 24 of whom were working mainly in animal welfare. The veterinarians were harmed by 6 food producers and 25 animal owners.

## 4. Discussion

Our findings showed that official veterinarians in Germany experienced high work demands but also strongly influenced their work. The relationship with supervisors/management differed, whereas the relationships with colleagues were good in most cases. Finally, more than one-third of the veterinarians reported experiences with threats of violence.

### 4.1. Study Design and Selection Bias

More than 800 official veterinarians participated in this study. However, as three-quarters of the target population did not respond, selection bias cannot be ruled out. The results of the comparison of the study population with the target population showed that women were more likely to participate. This phenomenon has also been observed in other studies [4,6,28]. The participation in an open online survey depends on the personal interest of the addressed persons. Women were maybe more likely to participate as they felt more addressed, because women are more likely to suffer from mental illnesses (gender gap in mental health [29]) and maybe also talk more openly about difficulties. Nevertheless, concerning cognitive and emotional demands, no differences were apparent between women and men. Different from other surveys among veterinarians [4,30,31], older veterinarians were not under-represented. The association of official veterinarians in Germany, which originally represented the official veterinarians employed as civil servants, distributed the link to our survey. This might be the reason why the participation from civil servants and thereby older persons was higher (Table 1). The results might thus overestimate the level of job security because civil servants were over-represented.

In general, the COPSOQ III is a well-established instrument used to assess different dimensions of psychosocial conditions among employees [25,26,27]. However, as the COPSOQ III is not a ready-to-use instrument, the surveys vary. Therefore, caution is warranted when comparing the results with those of other surveys, as dimensions with the same names might contain different items. We controlled for this problem by calculating the dimensions by following Lincke et al. [26]. We did not include all items of the GSV in our questionnaire to ensure the survey would not be too long. Therefore, the calculations for only a small proportion of dimensions were the same as those for the reference population. Due to the additional questions, survey completion required approximately 21 min to complete, a time span that should not be exceeded [32], as this might result in more dropouts and lower-quality answers [33].

### 4.2. Demands

Our results showed that official veterinarians in Germany experienced high job demands. A too-high workload for official veterinarians was also reported by Väärikkälä et al. [16]. Official veterinarians also reported experiencing higher emotional demands than the reference population (Table 3) and slightly higher demands than the employees working in healthcare, the social sector, and teaching and educating (mean = 69) [26]. These high demands might be caused by different circumstances: the official veterinary profession experiences a shortage of staff [24]. Due to the energy crisis and other factors, cities and districts are often in a precarious financial situation. Additionally, the workload presumably increased due to increases in the number of cases of animal neglect following inflation, new regulations such as the Animal Health Law, and the spread of African swine fever and avian influenza. Additionally, the increased workload could be an aftereffect of the COVID-19 pandemic. According to McKee et al. [34], approximately 50% of veterinarians in academic positions reported increased work as well as quantitative and emotional demands due to the COVID-19 pandemic.

A review by Stetina and Krouzecky [35] indicated a gender gap in mental health in veterinary medicine, as elsewhere: women veterinarians experience slightly higher demands [36], are more likely to use negative stress management practices [37], and suffer more often from suicidal ideation, which are more often related to the work situation [31]. In this study, the *Emotional* and *Cognitive Demands* dimensions did not vary among the genders or age groups. Even though different characteristics are assigned to the Baby Boomers, Generation X, and Generation Z [38], all generations of official veterinarians reported experiencing high emotional and cognitive demands. The results of our study contribute to preventing recriminations among genders and generations.

In contrast to these demographic factors, job characteristics significantly influenced the demands (Table 3). Welfare inspections, especially, were reported to be stressful [16]. In this study, veterinarians working mainly in animal welfare reported significantly higher emotional demands (Table 3). This finding is understandable as most of the conflicts, threats, and violence arose from situations involving animal owners. Moreover, veterinarians are confronted with both human and animal misery. Official veterinarians, as empathetic human beings, face challenges in dealing with human and animal suffering [39]. In the work of Devitt et al. [39], official veterinarians also reported an uncertainty: they felt unqualified to provide the necessary support to animal owners, e.g., in cases of farmers experiencing serious mental health problems. In contrast to persons working in, e.g., youth welfare services, knowledge regarding mental health and social support for a person in crisis is not yet always part of the education for official veterinarians in Germany. Here, courses and networks should be offered.

### 4.3. Influences and Possibilities for Development

Until now, research on the influences and possibilities for development and the control over working time of official veterinarians has been limited. In the work of Väärikkälä et al. [16], 70% of veterinarians reported working from home. In this study, the percentage was substantially lower (Table 1), even though our study was conducted after the COVID-19 pandemic, which increased the possibilities of working from home. The *Possibilities for Development* dimension consisted of two questions concerning the ability to use available expertise and to learn new things. The work of an official veterinarian is highly specialized, and regulations and conditions changing over time explain the relatively high scores (Table 2). The participants also rated their work as meaningful. Meaning and purpose are important drivers of work engagement, especially for Generation Z [40].

### 4.4. Social Relationships and Leadership

The relationships with colleagues were rated as good in most cases, whereas those with the supervisor and management differed among participants (Table 2). We obtained results similar to those of a survey of veterinarians employed in various fields of work [41]: whereas a good working atmosphere and good relationships with colleagues were mentioned the most often as positive aspects of work, the relationship with the employer/supervisor was reported as being sometimes difficult. In most studies on the stress and mental health of veterinarians, the relationships with colleagues and supervisors were not analyzed or reported [1]. Here, further research is needed because positive social relationships and support from supervisors and colleagues contribute to coping with stress: Väärikkälä et al. [16] found that support availability was associated with a higher work commitment, less feelings of loneliness, and even fewer sleeping disorders.

### 4.5. Job Security and Job Satisfaction

The veterinarians in this study felt secure in their positions probably due to the shortage of skilled labor and their long-term employment. This security is presumably a driver for this field of the profession, e.g., as long-term work contracts are rarely available in the academic field of veterinary medicine [41].

The overall job satisfaction reported in this study (Table 2, mean = 61) mirrors the job satisfaction of the reference population (mean = 63; [26]). Notably, the job satisfaction scale in the work of Lincke et al. [26] included three additional questions. Therefore, the comparability of these results is limited. In a recent study, the job satisfaction of veterinarians working in the noncurative sector was higher than that of those working in practice in Germany [41]. This finding can be explained by two factors: a higher salary and less time providing on-call services [42]. However, veterinarians might be less attracted by the salary for official services as salaries in the curative sector rise, which could be a reason for the shortage of younger official veterinarians.

### 4.6. Conflicts and Offensive Behavior

One of the motivations of this study was to assess the level of offensive behavior experienced by official veterinarians. In the work of Väärikkälä et al. [16], 88% of the participants experienced threatening situations during work in the previous 12 months. In our study, “only” 53% of official veterinarians working in animal welfare reported having been threatened. This difference might have multiple reasons: First, the definition of being threatened may vary among individuals and societies. Second, the wording or translation of the question might have biased the results. However, with an incidence of 30 physical attacks in 12 months, official veterinarians regularly experienced violence. Here, further research is needed concerning measures to prevent these events and to protect and support official veterinarians during and after their work. As an insight from this study, participants also assessed the availability and usefulness of certain measures (manuscript under preparation).

## 5. Conclusions

This is one of the first studies on the wellbeing of official veterinarians ensuring animal welfare, food safety, and control of infectious diseases and performing other tasks for veterinary public health. This study showed high job demand levels for official veterinarians in Germany. Moreover, those working mainly in animal welfare regularly experienced offensive behavior. Further insights are needed into how these demands influence job satisfaction and mental health. Additional approaches to support veterinarians during their work are necessary. Finally, further research is needed to explore similarities to and differences from the psychosocial situations of the official veterinarians in other countries.

Concerning the relationship with management/supervisors, further analyses including free-text answers or subsequent qualitative studies might provide further insight regarding the sources of problems and cooperation.

Positive organizational behavior (POB) focuses not on the weaknesses or mental illnesses of employees but, following the lead of positive psychology, on strengths and resources. Thereby, employees are not just prevented from experiencing mental illness but hopefully gain satisfaction, self-efficacy, and security within their jobs. The positive aspects of the work of official veterinarians in this study were the atmosphere and support from colleagues. Moreover, most participants experienced their work as meaningful and felt secure concerning their employment. These aspects should be highlighted to motivate veterinarians to work in this field. By reducing the demands and strengthening the positive aspects, more official veterinarians can hopefully be recruited and retained, thereby helping to ensure animal health, welfare, and food safety.

## Figures and Tables

**Figure 1 animals-14-01975-f001:**
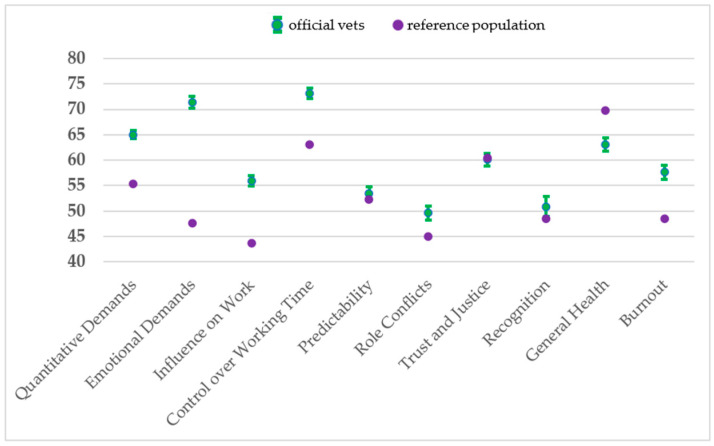
Means with 95% confidence intervals of scores for sociopsychological dimensions from a survey of official veterinarians in Germany (n: 813–838) compared with those of a reference population of German employees in different professions (Lincke et al. 2021: COPSOQ III in Germany: validation of a standard instrument to measure psychosocial factors at work [26]; n: 148,435–254,551); the dimensions included the following items: Quantitative Demands: WP1, WP2, QD2, QD3, CT5; Emotional Demands: EDX2, ED3; Influence at Work: INX1, IN3, IN4; CT: CT1, CT2; Predictability: PR1, PR2; CO: CO2, CO3, IT1; Trust and Justice: TM1, TMX2, JU1, JU4; Recognition: RE1; General Health: GH2*10; Burnout Symptoms: BO1, BO2, BO3; see Appendix A).

**Figure 2 animals-14-01975-f002:**
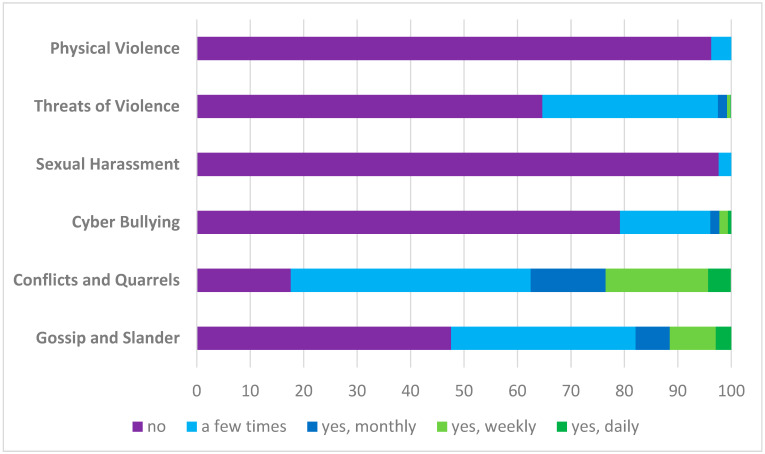
Experiences with concerning conflicts and offensive behavior in the last 12 months reported by official German veterinarians (*n* = 807) on a survey based on COPSOQ III.

**Table 1 animals-14-01975-t001:** Demographic information and job description from a survey assessing the psychosocial conditions of official veterinarians (*n* = 838).

	*n*	Percentage	Percentage in Target Population ^1^	Percentage in Reference Population ^2^
Gender				
Female	658	78.5	69.7	48.3
Male	179	21.4		51.7
Diverse	1	0.1		
Age				
<35 years	106	12.7		28.7
35–44 years	288	34.4		29.9
45–54 years	258	30.8		17.7
>54 years	186	22.2		13.0
Employment				
Civil servant	403	48.1	33.3	
Employed (temporary or tenure)	437	51.0	66.7	
Miscellaneous	8	1.0		
Working hours				
Part-time	274	32.9		26.1
Full-time	560	67.2		73.9
Level				
State	12	1.4		
Federal state	28	3.3		
City or province	798	95.2		
Main field of work				
Animal welfare	368	43.9		
Food safety	176	21.0		
Infectious animal diseases	140	16.7		
Ante- and postmortem inspection procedures (slaughterhouse)	64	7.6		
Other	87	10.4		
Work on weekends or public holidays (≥once/month)				
Yes	309	36.9		44.1
No	529	63.1		55.9
Work during evening/night (≥once/week)				
Yes	224	26.7		41.2
No	614	73.3		55.9
Possibility of working from home				
Yes	392	46.8		
No	446	53.2		
Work on the road/inspections				
Yes	748	89.3		
No	90	10.7		

^1^ Compared with all German official veterinarians (Bundestierärztekammer e.V. 2022 [11]). ^2^ Compared with a reference population of German employees in different professions (Lincke et al., 2021: COPSOQ III in Germany: validation of a standard instrument to measure psychosocial factors at work [26]; *n*: 148,435–254,551).

**Table 2 animals-14-01975-t002:** Descriptive analyses of psychosocial dimensions from survey of 838 German official veterinarians using COPSOQ III.

Dimension	Item ^1^	Positive Value	*n*	Mean	STD
DEMANDS					
Work Pace	WP1, WP2	low	838	67.8	18.3
Quantitative Demands	QD2, QD3	low	838	75.1	22.8
Cognitive Demands	CD1, CD2, CD4	low	838	81.7	11.6
Emotional Demands	ED1, EDX2, ED3	low	838	66.0	16.5
Demands for Hiding Emotions	HE2	low	838	64.4	20.7
Work Privacy Conflicts	WF2, WF3	low	838	51.7	23.8
INFLUENCE AND POSSIBILITIES					
Influence at Work	INX1, IN3, IN4, IN6	high	836	55.9	15.4
Possibilities for Development	PD2, PD3	high	838	73.8	17.2
Control over Working Time	CT1, CT2, CT4, CT5 *	high	837	63.6	13.9
Meaning of Work	MW1	high	838	73.3	20.5
SOCIAL RELATIONSHIPS AND LEADERSHIP					
Predictability	PR1, PR2	high	838	53.4	20.2
Recognition	RE1	high	823	50.9	28.2
Role Clarity	CL1, CL3	high	838	60.3	20.3
Role Conflicts	CO2, CO3	low	838	48.0	21.8
Illegitimate Tasks	IT1	low	838	52.6	22.1
Quality of Leadership	QL2, QL3, QL4	high	798	46.8	25.3
Social Support from Supervisor	SSX2	high	809	65.7	28.7
Social Support from Colleagues	SC1	high	822	79.6	20.4
Sense of Community at Work	SW1	high	823	79.5	17.2
JOB SECURITY AND SATISFACTION					
Job Insecurity	JI1, JI3	low	822	17.1	21.0
Insecurity over Working Conditions	IW1	low	821	21.7	24.8
Job Satisfaction	JS1, JS4, JS5, JSX	high	820	60.9	17.2
Commitment to Workplace	CW1, CW2, CWX3	high	838	55.5	20.0
Work Engagement	WE2	high	814	45.7	23.3
(MENTAL) HEALTH					
Self-rated Health	GH2	high	813	6.3	1.8
Sleeping Troubles	SL1	low	814	56.1	26.3
Burnout	BO1, BO2, BO3	low	814	57.6	20.7
Stress	ST1, ST3	low	814	60.6	21.8
Somatic Stress	SO1, SO2	low	814	29.4	22.0
Cognitive Stress	CS1, CS3	low	814	42.3	20.1
Depressive Symptoms	DS1, DS2, DS3, DS4	low	814	37.3	21.0
Self-Efficacy	SE2, SE4	high	814	66.7	19.7

* reversed scale. ^1^ See Appendix A or COPSOQ Guideline (https://www.copsoq-network.org/guidelines; [27] accessed on 17 October 2023); STD: standard deviation.

**Table 3 animals-14-01975-t003:** Differences in *Emotional Demands, Cognitive Demands,* and *Burnout Symptoms* dimensions according to gender, age group, and main field of work of 838 official German veterinarians.

	*Emotional Demands*						*Cognitive Demands*					*Burnout Symptoms*			
Category	*n*	Mean	LCI	UCI	*p*-Value	*n*	Mean	LCI	UCI	*p*-Value	*n*	Mean	LCI	UCI	*p*-Value
Sex															
Female	658	66.4	65.1	67.6	0.256	658	81.8	81.0	82.7	0.555	640	59.2	57.7	60.8	<0.001
Male	179	64.7	62.3	67.1		179	81.0	79.2	82.8		173	51.3	48.3	54.3	
Age															
<35 years	106	67.8	65.0	70.6	0.806	106	81.6	79.4	83.8	0.897	104	56.1	51.8	60.4	0.016
35–44 years	288	66.6	64.8	68.3		288	82.2	80.9	83.5		279	60.2	57.9	62.4	
45–54 years	258	65.3	63.1	67.5		258	81.4	79.9	82.8		253	58.9	55.9	61.0	
>54 years	186	65.3	62.9	67.7		186	81.2	79.5	83.0		178	53.1	50.0	56.3	
Main field of work															
Animal Welfare	106	70.7	69.3	72.2	<0.001	106	81.7	80.6	82.7	0.922	358	57.4	55.3	59.6	0.774
Else *	288	62.3	60.8	63.9		288	81.6	80.5	82.8		456	57.7	55.8	59.6	
Supervisor of other veterinarians															
Yes	348	68.0	66.3	69.6	0.006	348	84.3	83.2	85.4	<0.001	336	57.7	55.5	59.9	0.984
No	490	64.6	63.1	66.1		490	79.7	78.7	80.8		478	57.5	55.6	59.3	

* Includes infectious animal diseases, food safety, ante- and postmortem meat inspections, and miscellaneous.

## Data Availability

The data presented in this study are available on request from the corresponding author due to privacy reasons.

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
