# Peer review of "Survey on Psychosocial Conditions of Official Veterinarians in Germany: Comparison with Other Professions and Differences between Age Groups, Gender, and Workplace Characteristics"

_animals, 2024, doi:10.3390/ani14131975_

Round 1

Reviewer 1 Report

Comments and Suggestions for Authors

Introduction:

line 71 "were insulted"  do you mean attacked? injured? assaulted?

line 76:  "strong" should be strongly

Materials and Methods

line 104: Please explain what "not a ready--to-use questionnaire" means and exactly what modifications you made for your survey.  You can then refer back to this as you discuss the results and discussion of the results.

line 107:  What is GH1 and GH2?

line 117: "despite the open ended questions"  do you mean except the open-ended questions?

line 130:  what does implemented mean here?

line 143 "who stated not to work"  This is awkward wording and an example to English which needs work throughout the manuscript.   Other examples:  "stated to be"  could use reported "in average" should use on average  "leading position"  does this mean supervisors of other veterinarians?

Results

lines 179-181:  You might explain the difference between the target population and the reference population, and it would be useful to do this in the materials and methods section in detail.

line 186: "have leadership functions"  does this mean supervisory?

line 247: "I enjoy to tell others about my job"  This is awkward wording as an English translation.

lines 263-265: "the burnout assessment differed between gender and age groups..."  in what way did burnout differ by gender and age?

Page 9: Figure 1 please provide a key with the color that represents the veterinarians and the reference group. 

lines 291-293:  Please provide details about these questions in the materials and methods section.  What types of harm were asked about?  Who was involved in gossip/slander and about whom?

Discussion

line 310 "in very most cases" should be in most cases.

lines 316-317:  please clarify what you mean with this sentence.

line 325: awkward wording.

line 336: awkward wording.

line 348: "neglection" is not a word.

line 357: "suicidal ideas"  most often this is worded as suicidal thoughts or suicidal ideation.

line 361-362: What do you mean by recriminations between genders and generations? 

lines 370-372:  This is an important point, but it is not clear what you are trying to say based on how this is worded.

General comments:

The manuscript needs to be carefully edited for proper use of English.  The translation here is often awkward and needs to be addressed.  Examples of specific areas are provided in the comments above that can be used to assist in this.

The authors use the term "official" veterinarians throughout the manuscript but that is not a term most readers will be familiar with.  From reading the paper, I infer they are talking about government employees.  It would be helpful to clarify this.  It is not clear what the difference is between the group they call civil servants and the other official veterinarians.  Please provide some more information for readers about the study population. 

Comments on the Quality of English Language

The English needs to be carefully reviewed for proper translation.  I have noted some specific things that need to be corrected above but did not mark every instance where editing is needed.  

Author Response

Thank you for your review and your helpful comments. As most of your comments refer to language, we sent the manuscript to an Editing Service.

Introduction:

line 71 "were insulted"  do you mean attacked? injured? assaulted?

We meant they were injured. We replaced the word (see l. 89)

line 76:  "strong" should be strongly

We changed the word (l. 94)

Materials and Methods

line 104: Please explain what "not a ready--to-use questionnaire" means and exactly what modifications you made for your survey.  You can then refer back to this as you discuss the results and discussion of the results.

We changed the structure of this paragraph to make it clearer (see ll. 128-142)

line 107:  What is GH1 and GH2?

These are two items of the COPSOQ. We deleted the information at that point as we think it is more confusing than helpful. The information is still traceable in Table 1 and Supplementary Table S1.

line 117: "despite the open ended questions"  do you mean except the open-ended questions?

Yes, you are right. Thank you. We changed the word (l. 152).

line 130:  what does implemented mean here?

LimeSurvey is a cloud-based software for online survey. We transferred our questionnaire to LimeSurvey in order to conduct the survey.

line 143 "who stated not to work"  This is awkward wording and an example to English which needs work throughout the manuscript.   Other examples:  "stated to be"  could use reported "in average" should use on average  "leading position"  does this mean supervisors of other veterinarians?

We changed “stated to” in “reported to” or “answered” or changed the phrasing otherwise. Moreover, we changed “in average” to “on average” (l. 266) and changed the phrasing in l. 236.

Results

lines 179-181:  You might explain the difference between the target population and the reference population, and it would be useful to do this in the materials and methods section in detail.

Thank you for that helpful comment. We included the information in l. 201-204.

line 186: "have leadership functions"  does this mean supervisory?

Yes, we changed the sentence to “About 41% (n=348) of the participants reported performing a supervisory function for other official veterinarians.” (l. 235-236)

line 247: "I enjoy to tell others about my job"  This is awkward wording as an English translation.

We changed the translation to the wording of the English version of the COPSOQ: “Do you enjoy telling others about your place of work?” (l. 305)

lines 263-265: "the burnout assessment differed between gender and age groups..."  in what way did burnout differ by gender and age?

We added the information included in Table 3 to this paragraph: “Women reported a higher level of burnout symptoms than men, and persons in the age group 35-44 years reported a higher level of burnout than those older than 54 years.” (ll. 326-328)

Page 9: Figure 1 please provide a key with the color that represents the veterinarians and the reference group. 

Sorry, I do not understand. You mean a legend? This is provided on top of the figure.

lines 291-293:  Please provide details about these questions in the materials and methods section.  What types of harm were asked about?  Who was involved in gossip/slander and about whom?

Thank you for this helpful comment. We added the information in ll. 145-149: ”Participants were asked about their experience in the last 12 months concerning gossip and slander, conflicts and quarrels, cyber bullying, sexual harassment, threats of violence and physical violence. If a person reported that they experienced such behavior, they were asked by whom (owner of animals, food producer, coworker, supervisor, or someone else).” We only asked who behaved offensive not the cause or about whom.

Discussion

line 310 "in very most cases" should be in most cases.

Changed (l. 378)

lines 316-317:  please clarify what you mean with this sentence.

There is a gender gap in mental health: women are more likely to suffer from (certain) mental disorders than men and are also rather willing to admit difficulties in mental health. Both factors might have attracted women to participate as the voluntary participation in online survey depend on the personal interests of a person. However, in most studies – also on other topics in veterinary medicine – more women and less men participated than expected. We explained the point in more detail (ll. 385-389)

line 325: awkward wording.

You are right. We deleted the last part of the sentence.

line 336: awkward wording.

We deleted again a part of the sentence to make it clearer.

line 348: "neglection" is not a word.

We changed the word to neglect (l. 424).

line 357: "suicidal ideas"  most often this is worded as suicidal thoughts or suicidal ideation.

You are right, it should have been suicidal ideation (l. 433)

line 361-362: What do you mean by recriminations between genders and generations? 

To our experience, older (often male) veterinarians sometimes tend to accuse younger (often female) veterinarians of being not enough efficient, powerful or engaged. On the other hand, younger veterinarians sometimes blame the older generations for being not open-minded. However, this is just our opinion… to our knowledge there are no studies concerning these generation conflicts.

lines 370-372:  This is an important point, but it is not clear what you are trying to say based on how this is worded.

We changed the wording to: “In Devitt et al., official veterinarians also reported an uncertainty: they felt unqualified to provide the necessary support to animal owners, e.g., in cases of farmers experiencing serious mental health problems.” (ll. 448-451)

General comments:

The manuscript needs to be carefully edited for proper use of English.  The translation here is often awkward and needs to be addressed.  Examples of specific areas are provided in the comments above that can be used to assist in this.

Thank you for your comments. We sent the manuscript to an editing service.

The authors use the term "official" veterinarians throughout the manuscript but that is not a term most readers will be familiar with.  From reading the paper, I infer they are talking about government employees.  It would be helpful to clarify this.  It is not clear what the difference is between the group they call civil servants and the other official veterinarians.  Please provide some more information for readers about the study population. 

We added a definition in the abstract (ll.25-26), results section (220-222) and conclusion (ll.515-517). We thought about using the term “government veterinarians”, but rejected the idea as it can also be misleading. We also have veterinarians in the government in some ministries. These were part of our target group, but most participants worked for districts or cities. We also decided against the term “public veterinarians” as this term is used also for veterinarians providing Animal Health Services funded by the government.

Reviewer 2 Report

Comments and Suggestions for Authors

I enjoyed reading this paper and felt the authors presented some very interesting findings in relation to work stress for official veterinarians. I was very interested in the findings of high emotional demands and the finding of high rate of threats of violence. As the authors discuss this has important implications for programs and policies to both prevent such occurrences and also support workers as they are confronted with difficult situations.

My main suggestion for improvements is to discuss the implications of the findings for an international audience. In the introduction research from many countries was cited. The introduction then seemed to narrow to a gap within the German context. I assume there is not much research about official veterinarians in the international literature and perhaps this point could be made as well. Is the distinction between practice and official veterinarians similar in other countries and thus are these findings relevant to an international audience. Is there any literature on official veterinarians in other countries?

Similar in the discussion, I would have found it interesting in the discussion if the specific location of the research cited could have been provided. I assume there maybe different policies and government institutions that regulate this type of work in different countries. Thus it may be interesting to explore whether the rates of these types of issues are different. And if there are differences this may help in designing appropriate policy and program solutions. If there is little direct research for which to compare then this research gap could be explained more explicitly.

The conclusion did a good job of highlighting areas of importance to concentrate on within the context of this study: official veterinarians in Germany. Providing some thoughts to direct research internationally may also be of value in the conclusion.

Comments on the Quality of English Language

Some editing of English required in places such as use of present and past tense.

Author Response

Dear reviewer,

Thank you for reviewing the manuscript and your positive feedback. To our knowledge, except the study by Väärikkälä et al. from Finland, this is the first study exploring the mental wellbeing of official veterinarians. Therefore, you are right – there is definitely a gap in research on this topic. We added some information and emphasized the need of further studies (see ll. 522-524).

We also thought about adding some comparisons to results of other COPSOQ surveys from other countries, but then rejected this idea. All studies focused on other professions in other countries in other languages. Therefore, differences in means can be caused by either the profession or the country/ society or the translation and we cannot distinguish between them.

Reviewer 3 Report

Comments and Suggestions for Authors

Dear authors,

I appreciate the opportunity to review your manuscript. Your work contributes significantly to the analysis and understanding of work demands, particularly for professionals continuously exposed to emotionally distressing situations, such as those in animal welfare. I found your manuscript highly engaging.

I would like to offer some suggestions to enhance certain aspects of your work.

Theoretical Considerations:

The Introduction could benefit from greater depth. Specifically, regarding the tasks typically performed by official veterinarians and their potential impact on mental health. Given the variety of tasks, each with distinct emotional impacts, it would be helpful to differentiate them based on their effects on individuals.

Methodological Considerations:

Two methodological aspects need attention and possible revision:

1. Comparative Analysis: The manuscript presents comparative results between official veterinarians and employees with different professions, implying an inferential analysis for one-sample (e.g., one-sample t-test or signed rank Wilcoxon test). It is important to include the relevant statistical notation to support these results, both in the manuscript text and in Figure 1.

2. Statistical Test Selection: The rationale for using the Kruskal-Wallis test for statistical analysis is unclear, particularly regarding which assumptions were violated. If comparing two groups (e.g., gender), a Mann-Whitney test would be more appropriate. Additionally, when reporting Kruskal-Wallis test results, the values reported should be the medians (mean ranks), not the means.

Thank you once again for the opportunity to read your manuscript.

Sincerely,

Author Response

Dear authors,

I appreciate the opportunity to review your manuscript. Your work contributes significantly to the analysis and understanding of work demands, particularly for professionals continuously exposed to emotionally distressing situations, such as those in animal welfare. I found your manuscript highly engaging.

I would like to offer some suggestions to enhance certain aspects of your work.

Dear reviewer, thank you for reviewing our manuscript and your kind feedback!

Theoretical Considerations:

The Introduction could benefit from greater depth. Specifically, regarding the tasks typically performed by official veterinarians and their potential impact on mental health. Given the variety of tasks, each with distinct emotional impacts, it would be helpful to differentiate them based on their effects on individuals.

Thank you for your suggestion. We added some information in ll. 78-86 and 96-98 concerning the everyday work of official veterinarians.

Methodological Considerations:

Two methodological aspects need attention and possible revision:

  1. Comparative Analysis: The manuscript presents comparative results between official veterinarians and employees with different professions, implying an inferential analysis for one-sample (e.g., one-sample t-test or signed rank Wilcoxon test). It is important to include the relevant statistical notation to support these results, both in the manuscript text and in Figure 1.

We do not have the data of the reference group but refer only to the results published open access. The data is not available for free as it belongs to a private institute. We added this information in ll. 203-204. Moreover, we think that inferential statistics are not necessary as the confidence intervals provide sufficient information about significant differences between these two groups as the intervals do not overlap.

  1. Statistical Test Selection: The rationale for using the Kruskal-Wallis test for statistical analysis is unclear, particularly regarding which assumptions were violated. If comparing two groups (e.g., gender), a Mann-Whitney test would be more appropriate. Additionally, when reporting Kruskal-Wallis test results, the values reported should be the medians (mean ranks), not the means.

You are right: statistical tests and descriptive analyses do not really fit. The data is not (always) normally distributed as some data was not really continuous (see also ll. 204-208). However, we chose to report means and confidence intervals in most cases to keep the results comparable to the reference population. We then decided to choose a test for non-parametric data. These tests fit for non-normally distributed data but also provide valid results for normally distributed data. In case of normal distribution, p-values might be higher than the corresponding test for normally distributed data. So, our procedures were more conservative but also revealed significant differences as the sample size was high.

Based on your suggestion, we calculated the differences between those variables with two categories (gender, field of work, supervisor for other veterinarians) using the Mann-Whitney-test and replaced the p-values and added the corresponding information in the material and methods section. Interestingly, the p-values for differences between genders did not change.

Thank you once again for the opportunity to read your manuscript.